# Highly Dispersed CeO_x_ Hybrid Nanoparticles for Perfluorinated Sulfonic Acid Ionomer–Poly(tetrafluoethylene) Reinforced Membranes with Improved Service Life

**DOI:** 10.3390/membranes11020143

**Published:** 2021-02-18

**Authors:** Juhee Ahn, Mobina Irshad Ali, Jun Hyun Lim, Yejun Park, In Kee Park, Denis Duchesne, Lisa Chen, Juyoung Kim, Chang Hyun Lee

**Affiliations:** 1Department of Energy Engineering, Dankook University, Cheonan 31116, Korea; jhahn9212@gmail.com (J.A.); iyj2368@naver.com (J.H.L.); kddyejun@gmail.com (Y.P.); inkee0149@gmail.com (I.K.P.); 2Department of Advanced Materials Engineering, Kangwon University, Samcheok 25913, Korea; mobinachemist@gmail.com; 33M Advanced Materials Division, 3M Center, St. Paul, MN 55144, USA; dduchesne@mmm.com (D.D.); lpchen@mmm.com (L.C.)

**Keywords:** radical scavenger, service life, polymer electrolyte membrane

## Abstract

CeO_x_ hybrid nanoparticles were synthesized and evaluated for use as radical scavengers, in place of commercially available Ce(NO_3_)_3_ and CeO_2_ nanoparticles, to avoid deterioration of the initial electrochemical performance and/or spontaneous aggregation/precipitation issues encountered in polymer electrolyte membranes. When CeO_x_ hybrid nanoparticles were used for membrane formation, the resulting membranes exhibited improved proton conductivity (improvement level = 2–15% at 30–90 °C), and thereby electrochemical single cell performance, because the –OH groups on the hybrid nanoparticles acted as proton conductors. In spite of a small amount (i.e., 1.7 mg/cm^3^) of introduction, their antioxidant effect was sufficient enough to alleviate the radical-induced decomposition of perfluorinated sulfonic acid ionomer under a Fenton test condition and to extend the chemical durability of the resulting reinforced membranes under fuel cell operating conditions.

## 1. Introduction

Polymer electrolyte membrane fuel cells (PEMFCs) are energy conversion systems to directly convert the chemical energy of hydrogen into electricity via an electrocatalytic redox reaction [1,2]. A current technical issue in the development of PEMFCs relates to how long their service life can be extended while maintaining the electrochemical performance as high as possible [3,4,5,6]. One of the key components to determine these tough requirements is polymer electrolyte membrane (PEM).

The state-of-the-art PEM materials are perfluorinated sulfonic acid (PFSA) ionomers composed of a hydrophobic poly(tetrafluoroethylene) (PTFE) main chain with excellent chemical resistance and perfluorinated side chains with a hydrophilic sulfonic acid (–SO_3_^−^) group at each terminal. Their architectural features give rise to developed hydrophobic–hydrophilic microphase-separated morphologies [7,8,9] when the materials are used for PEM formation. The hydrophilic channels formed within their morphologies by the self-assembly of their hydrophilic moieties are used as proton transport pathways. In addition to the well-defined morphologies, electronegative fluorine atoms, particularly those adjacent to –SO_3_^−^H^+^ groups, cause protons to be easily released from –SO_3_^−^ groups and contribute to fast proton conduction [7,8]. It is, however, difficult to apply PFSA materials in the freestanding membrane form for highly durable fuel cell applications owing to their technical limitations, which include low dimensional stability during repeated wet and dry cycles, weak mechanical toughness, and low hydrogen barrier properties [10,11]. These weaknesses are more serious when thinner membranes (e.g., <18 µm) are applied for reducing the stack volume.

One plausible way to overcome the shortcomings of these freestanding membranes is to prepare reinforced membranes. Representative reinforced membranes are PFSA–PTFE reinforced membranes, which are fabricated by filling the pore of PTFE support films with PFSA ionomers dispersed in aqueous alcohol media (e.g., propanol and/or ethanol) before evaporating off these solvent mixtures while preventing the formation of defects or unfilled pores [12,13,14,15]. The PTFE support films prevent PFSA matrices in their pores from being excessively swollen, particularly in the planar direction, and lead to enhanced mechanical durability and proton conductivity in the resulting reinforced membranes [12,15]. The hydrogen permeability in PTFEs is lower than that in PFSA materials, which helps in improving hydrogen barrier properties in the PFSA–PTFE membranes. Furthermore, thin PFSA–PTFE reinforced membranes have other merits, such as low areal resistance to proton transport, easy water absorption even under low humidity conditions, and relatively low material cost [16].

It is, however, necessary to further improve their resistance to chemical degradation of the PFSA matrices in the reinforced membranes, which occurs owing to the oxidative attack of aggressive free radicals, such as hydroxyl radical (·OH), hydrogen radical (·H), and oxygen radical (·O_2_) [17,18,19]. The C–S bonds and swivel –O– groups in the PFSA materials are vulnerable to the attack of ·OH and ·H radicals, respectively [20]. These radicals have been known to be generated as a result of incomplete oxygen reduction reaction and/or hydrogen crossover through PEM. The chemical decomposition of the PFSA matrices is accompanied by increased hydrogen crossover and irreversible losses in their electrochemical potentials, including open-circuit voltage (OCV) [11]. Here, OCV is defined as the maximum electromotive force at an open circuit (current density = 0 A/cm^2^) without any ohmic load [21].

The chemical damage can be retarded by incorporating radical scavengers [11,22]. The radical scavengers are classified into ion-type and oxide-type. Representative ion-type radical scavengers are multivalent metallic cations, such as Ce^3+^ and Mn^2+^. These radical scavengers are generally provided in the solidified salt forms with a variety of polyanions (e.g., cerium (III) nitrate and Ce(NO_3_)_3_). These salts are easily dissolved in water or aqueous alcohol media in most cases and dissociate into the metallic cations and their counter anions. When added into PFSA dispersion for reinforced membrane formation, the metallic cations are exchanged with some H^+^ ions coordinated with –SO_3_^−^ groups of the PFSA ionomers and form ionically crosslinked complexes with the PFSA chains (e.g., –SO^3−^–Mn^2+^–^−3^OS–) owing to their multivalent characteristics. Consequently, the relative content of free –SO^3−^ groups related to the easy release of protons is inevitably reduced, which results in the loss of proton conduction capability in the corresponding membranes due to the combination of the mobility reduction effect of ionomer chains. Thus, a minimum content (e.g., <2.5 mg/cm^3^ in a PFSA matrix [23] or <3.8 µg/cm^2^ on the basis of a reinforced membrane with a thickness of 15 µm) has been introduced that can prevent radical scavengers from being aggregated and/or precipitated in the dispersion state and can minimize the decrease in proton conductivity in the membrane state at the same time. In other words, it means that the radical scavengers in the salt form cannot be incorporated beyond the limit, despite the possibility that the chemical durability may be further improved when more radical scavengers are added.

On the other hand, commonly used oxide-type radical scavengers include CeO_2_ [5,24], MnO_2_ [25], boehmite (Al_2_O_3_ [22]), and Ag/SiO_2_ [26]. The inorganic oxides are free from the formation of ionic complexes with –SO^3−^ groups in the PFSA matrices or the rigid crosslinked networks and thus do not induce reduced proton conductivity. Rather, the oxide-type radical scavengers can help in fastening proton conduction through the resulting reinforced membranes below a certain content because of the presence of –OH groups acting as additional acids when exposed to PEM materials as superacids (<pH 1, [27]). On the contrary, when more than a certain amount is added, the inorganic oxides are easily aggregated and heterogeneously dispersed [28,29]. This leads to localized mechanical failures and defects, which are followed by membrane brittleness and lower hydrogen barrier properties in the resulting membranes. This phenomenon is conspicuously observed as the average particle size of inorganic oxides decreases and/or inorganic oxides with high density values are used.

In this study, a new strategy to take the merits of both ion-type and oxide-type radical scavengers is addressed for obtaining effective radical scavengers to extend the lifetime without a loss in electrochemical performance. This strategy is based on the in situ chemical processes to convert cerium ions introduced into an aqueous alcohol mixture containing PFSA ionomers used as dispersants to CeO_x_ hybrid nanoparticles, maintaining high dispersibility for a long period of time. The objective of this study is to suggest a powerful radical scavenger preparation method. Another purpose is to disclose how the prepared hybrid nanoparticles affect PEM properties, including the electrochemical performance. Finally, their contribution to the chemical durability of the resulting reinforced membranes in the membrane–electrode assembly (MEA) state under a continuous OCV test condition is verified.

## 2. Materials and Methods

### 2.1. Materials

A commercially available porous PTFE support film (AX17, Donaldson, Bloomington, MN, USA) was used to make PFSA–PTFE reinforced membranes. 3M PFSA ionomer with an equivalent weight (EW) of 725 g/mol of –SO_3_^−^H^+^ (3M 725EW) was used as the PFSA material filled into the pores of the PTFE support film. N-propanol (CH_3_CH_2_CH_2_OH, anhydrous, purity = 99.7%), sulfuric acid (H_2_SO_4_, purity = 95.0–98.0%), hydrochloric acid (HCl, purity = 37%), and ammonium hydroxide (28% NH_3_ in water) were purchased from Sigma-Aldrich (St Louis, MO, USA) and used as received without any further purification. Cerium(III) nitrate hexahydrate (Ce(NO_3_)_3_.6H_2_O, 99%) was purchased from Alfa Aesar (Haverhill, MA, USA) and used as a CeO_x_ precursor.

### 2.2. Synthesis of CeO_x_ Hybrid Nanoparticles

Prior to the formation of CeO_x_ hybrid nanoparticles, PFSA dispersion with an ionomer concentration of 5 wt % was prepared by adding 3M 725EW ionomer in the powder state into an aqueous N-propanol mixture (N-propanol/water = 55:45 by weight ratio) and stirring the resulting solution at ambient temperature for 48 h. Then, 0.75 g of Ce(NO_3_)_3_.6H_2_O was added to 15 g of the PFSA dispersion being stirred. Immediately after complete dissolution of the CeO_x_ precursor, 0.51 g of 37% HCl was slowly added to the mixed solution at 25 °C and heated up to 60 °C. After hydrolysis reaction for 24 h, a transparent dispersion without color was obtained. The resulting dispersion was denoted as CeO_x__A dispersion. In addition, 0.156 g of 28% ammonium hydroxide was slowly added to the CeO_x__A dispersion to change its pH to a basic level for hydrolysis and condensation. After t base treatment at 60 °C for 24 h, a yellowish reaction product, denoted as CeO_x__B dispersion, was obtained.

### 2.3. Fabrication of PFSA–PTFE Reinforced Membranes

For PFSA–PTFE membrane formation, a 3M 725EW dispersion with an ionomer concentration of 12 wt % was prepared with the same method and solvent composition as the 5 wt % PFSA dispersion used for the synthesis of CeO_x_ hybrid nanoparticles. Each CeO_x_ dispersion was added to the relatively viscous 3M 725EW dispersion and mechanically mixed at room temperature for 2 h. Here, the loading content of cerium element was fixed at 2.5 mg per 3M 725EW ionomer volume (cm^3^) (in other words, 3.8 µg/cm^2^ on the basis of the reinforced membranes with thickness of 15 µm), regardless of which Ce type was used. After sonication for 2 h as a degassing step to remove the remaining microbubbles, each 3M 725EW dispersion containing CeO_x_ hybrid nanoparticles was used as a coating solution to prepare PFSA–PTFE reinforced membranes. After the coating step, all the PFSA–PTFE reinforced membranes were solidified at 85 °C for 12 h and thermally treated at 190 °C for 12 min at convectional atmosphere. The resultant PFSA–PTFE reinforced membranes were named as CeO_x__A PFM and CeO_x__B PFM depending on the type of CeO_x_ dispersion used. For comparison, a PFSA–PTFE reinforced membrane without the use of any CeO_x_ dispersions was also prepared and named as PFM. Finally, CeO_x__A PFM and CeO_x__B PFM were treated in 0.5 M H_2_SO_4_ for 2 h and boiling deionized water for another 2 h (method II [22,30,31,32,33,34]). The acid treatment has been widely used to remove metallic cations existing in the membranes by exchanging the cations that participate in the ionic bonding with –SO_3_^−^ groups in the PFSA matrices with protons, while the boiling water treatment was conducted to wash off excessive H_2_SO_4_ molecules from the membranes. After these treatments, CeO_x__A PFM and CeO_x__B PFM were converted into H–CeO_x__A PFM and H–CeO_x__B PFM, respectively. For comparison, Ce(NO_3_)_3_ PFM with the same cerium element content was also prepared. All the membranes had the same thickness value of 15 ± 1 µm.

### 2.4. Characterization

Dynamic light scattering (DLS; Zetasizer ZEM600, Nano ZSP, Malvern Instruments, Malvern, U.K) was used to measure the average size of CeO_x_ hybrid nanoparticles. The concentration of Ce^3+^ ion was quantitatively determined using an inductively coupled plasma optical emission spectroscopy system (ICP-OES; 5110 VDV, 27 MHz, Agilent Technologies, Santa Clara, CA, USA). Field emission scanning electron microscopy (FE-SEM; JEOL JSM-6701F/X-Max, New England, MA, USA) together with energy-dispersive X-ray (EDX) analysis was used to confirm the formation of CeO_x_ hybrid nanoparticles and to investigate their microstructure.

The ohmic resistance (*R*) of the PFSA–PTFE reinforced membranes (dimension of each membrane coupon = 1 × 4 cm^2^) was measured in deionized water from 30 to 90 °C using a four-point probe alternating current (AC) impedance spectroscopy with a home-made test cell connected with a potentiostat (Model VSP, BioLogic, Seyssinet-Pariset, France) [35]. The measurement was carried out with an amplitude of 100 µA in the frequency range of 100 mHz to 100 kHz. The measured ohmic resistance was converted to proton conductivity (*σ*, S/cm) using Equation (1):(1)$$\sigma =\frac{L}{RS}$$
where *L* and *S* mean the distance between the reference electrodes and the cross-sectional area of the membrane coupon in the in-plane direction, respectively.

The hydrogen permeability (*P*, Barrer or 10^10^ cm^3^_stp_ cm/(s cm^2^ cm Hg)) of the reinforced membranes was obtained as a function of temperature using Equation (2) on the basis of the time-lag method [36]:(2)$$P\;\left(Barrer\right)=\frac{V_{b}l}{p_{a}ART}\;\times \;\frac{dp_{b}}{dt}$$
where *V_b_* (cm^3^) is the volume of the bottom chamber, *l* (cm) is the membrane thickness, *A* (cm^2^) is the surface area of the membrane exposed to hydrogen gas, *T* (°K) is absolute temperature, *p_a_* (cm Hg) is the pressure of the top chamber, *R* (6236.367 cm Hg cm^3^/mol °K) is the gas constant, and *dp_b_/dt* is the rate of change of the pressure in the bottom chamber as a function of time measured in the linear part of the pressure–time curve (cm Hg/s).

The chemical resistance to oxidative radicals was evaluated by measuring the concentration of fluoride ions released from the reinforced membrane coupon (dimension = 2 × 2 cm^2^) under a Fenton test condition. For this, the test solution was prepared at 25 °C using 30 wt % H_2_O_2_ and 0.1 wt % ferrous ammonium sulfate. After duration in the Fenton solution at 80 °C for 2 h, the resulting solution was mixed with total ionic strength adjustment buffer II (TISAB^®^ II; ThermoFisher Scientific, Waltham, MA, USA) in the volume ratio of 1 to 1. The concentration of the fluoride ions in the mixed solution was measured using a pH/ion-selective electrode meter (Orion Star A214, ThermoFisher Scientific, Waltham, MA, USA) equipped with the ion selective electrode (Model 9609BNWP).

The electrochemical single cell performance was obtained at 65 °C under different humidity conditions (i.e., 100% and 50% relative humidity (RH)) using a test cell (PEM025-01, CNL Energy, Republic of Korea) with electrodes of 25 cm^2^ active area each. The feed gases were supplied maintaining the stoichiometric ratio of hydrogen to air of 1.5 to 2. All the MEAs were fabricated via a decal coating. The electrode slurry for the decal electrode formation was made by mixing 5 wt % Nafion dispersion (D521, Chemours, Wilmington, DE, USA) and 40 wt % Pt/C catalyst (3006Pt, VINATech, Republic of Korea). Each MEA was obtained after hot-pressing of the decal electrodes located on both sides of the reinforced membrane under a pressure of 10 MPa at 130 °C for 5 min. Here, the loading contents of Pt and Nafion binder in the electrodes were 0.4 and 0.25 mg/cm^2^, respectively.

The chemical durability evaluation, called the OCV holding test, was accomplished at 90 °C and 30% RH following a USA Department of Energy protocol [37]. The durability test was continuously carried out until the measured OCV value was reduced to a level higher than 80% of the initial OCV value or the hydrogen crossover electrochemically measured via the linear sweep voltammetry (LSV) reached up to a level higher than 15 mA/cm^2^. Here, the LSV test was conducted at 65 °C and 50% RH under N_2_ atmosphere.

## 3. Results and Discussion

Ce(NO_3_)_3_.6H_2_O is converted to CeO_x_ nanoparticles according to the following chemical reactions shown in Equations (3)−(5) [38,39,40]. In the presence of HCl as an acid catalyst, Ce(NO_3_)_3_ undergoes hydrolysis via the chemical reaction in Equation (3) and is transformed to Ce(OH)_3_. When Ce(OH)_3_ is exposed to aqueous NH_4_OH, Ce(OH)_4_ is able to be formed as a result of the condensation reaction in Equation (4). Ce(OH)_4_ can also be altered to CeO_2_ through an additional condensation reaction in Equation (5).

Even though both Ce(OH)_3_ and Ce(OH)_4_ have been conventionally called ceria (CeO_2_), their chemical identities, such as oxidation number and molecular weight, are somewhat different from those of CeO_2_. Because Ce(OH)_3_ and Ce(OH)_4_ can be chemically converted to CeO_2_ under a certain condition (e.g., excess hydrogen peroxide or alkaline condition [41]) and even CeO_2_ has hydroxyl (–OH) groups obtained as a result of the oxidation of its surface, it is very difficult to chemically or physically isolate these metallic oxides from each other. Ce(OH)_3_, Ce(OH)_4_, and CeO_2_ are known to give rise to antioxidant effect [41,42] to prevent the chemical degradation of PFSA ionomers to radical attacks. For this reason, all the cerium oxide derivatives will be collectively referred to as CeO_x_ in this study. CeO_x_ is present in the form of an insoluble nanoparticle composed of cerium and oxygen in water or the aqueous N-propanol mixture. The nanoparticles are spontaneously aggregated in the media and easily precipitated during hydrolysis–condensation reaction owing to their high density values (e.g., density of CeO_2_ = 7.215 g/cm^3^, [43]). Thus, this characteristic makes it difficult to homogeneously distribute CeO_x_ nanoparticles in the aqueous media used for membrane formation.
2Ce(NO_3_)_3_·6H_2_O + conc. HCl → 2Ce(OH)_3_ + 6HNO_3_ + H^+^ + Cl^−^(3)
2Ce(OH)_3_ + 2NH_4_OH + 1/2O_2_ → 2Ce(OH)_4_ + 2NH_3_ + H_2_O(4)
Ce(OH)_4_ → CeO_2_ + 2H_2_O(5)

The difficulty in the homogeneous distribution of CeO_x_ nanoparticles was solved using 3M 725EW ionomer as a polymeric dispersant. In the 3M 725EW dispersion used for hybrid formation, the ionomer was evenly distributed in the form of polymeric nanoparticles, maintaining an average size of 2.2 nm. When Ce(NO_3_)_3_.6H_2_O was dissolved in the aqueous dispersion, positively charged Ce^3+^ ions formed ionic bonding with negatively charged –SO_3_^−^ groups on the ionomer particles. In the dispersion obtained immediately after the Ce(NO_3_)_3_ dissolution (called fresh Ce(NO_3_)_3_ dispersion), the size of the polymeric nanoparticles interacting with multivalent Ce^3+^ ions increased slightly to 2.8 nm, indicating the formation of their ionic complexes. Within 12 h, the ionic complexes, however, became large enough to be observed with the naked eye (Table 1). This aggregation issue was avoided by adding a small amount of HCl catalyst into the fresh Ce(NO_3_)_3_ dispersion. Simultaneously, the HCl addition caused the chemical conversion from Ce(NO_3_)_3_ to Ce(OH)_3_ within the complexes. Consequently, this transformation was accompanied by the formation of Ce(OH)_3_–3M ionomer hybrid nanoparticles. The yield of this conversion reaction was calculated by measuring the change in Ce^3+^ ion concentration during the acid-catalyzed reaction based on ICP-OES. The initial concentration of Ce^3+^ ions in the dispersion gradually decreased with the reaction time. The difference between the initial concentration and the concentration measured after a certain period of time corresponded to the concentration of cerium element converted to CeO_x_. After 24 h, the conversion yield reached 34.2 mol %. This means that 65.8 mol % of the initial Ce^3+^ ions still existed in the unreacted state. Interestingly, the resulting CeO_x__A hybrid nanoparticles maintained high dispersion stability in the aqueous mixture without big changes in their average particle size even after a long period of more than six months. The formation of stable hybrid nanoparticles in the acidic atmosphere is highly related to the CeO_x_ dispersion characteristics depending on the pH value of the dispersion media. Note that CeO_x_ nanoparticles are stably dispersed in a pH level lower than their isoelectric point (IEP; IEP of CeO_x_ = pH 7–8 [44,45]) because the electrostatic repulsion between the nanoparticles with positive surface charges prevent them from being aggregated [46,47]. Furthermore, the strengthened electrostatic attraction between the Ce^3+^ ions in insoluble Ce(OH)_3_ and the –SO_3_^−^ groups of 3M 725EW ionomer may help in retaining a stable dispersed phase. It is also believed that this high dispersion characteristics of the CeO_x_ hybrid nanoparticles would be kept even when HCl catalyst is removed because the 3M 725 ionomer in contact with the CeO_x_ nanoparticles in the hybrid state provides an acidic atmosphere as superacid [27].

On the other hand, the ammonium hydroxide treatment of CeO_x__A dispersion caused changes in both the average size of the hybrid nanoparticles distributed in the aqueous N-propanol mixture and the CeO_x_ conversion yield. The average size of CeO_x__B hybrid nanoparticles was larger than that of CeO_x__A hybrid nanoparticles. This phenomenon might have come from the aggregation of CeO_x_ nanoparticles formed during the conversion reaction in Equation (5). It might also be due to the relatively high swelling of 3M 735EW ionomer in the hybrid nanoparticles under an aqueous weak base condition rather than under a strong acid condition. Even though unreacted Ce^3+^ ions still existed in the CeO_x__B dispersions obtained after the base treatment, their relative portion was considerably reduced after conversion for 24 h. This means that the basic treatment was effective in improving the conversion yield to CeO_x_. Meanwhile, the stability of CeO_x__B dispersion was reduced to about two months. After that period, a small amount of the particles aggregated and began to be precipitate as CeO_x_ nanoparticles are somewhat unstable in basic aqueous media [46,47].

Figure 1 shows the field-emission scanning electron micrographs of CeO_x__A and CeO_x__B hybrid nanoparticles containing 3M 725EW ionomer. Here, each sample was prepared with the same fabrication history used to make the PFSA–PTFE reinforced membranes, except for the use of a porous PTFE support film. The relatively high concentration of cerium to 3M 725EW ionomer (i.e., 1 to 1 weight ratio) in each dispersion induced the morphologies where the resulting hybrid nanoparticles were aggregated. The aggregation pattern was dissimilar. The agglomerates in the CeO_x__A matrix (Figure 1a) were small but uniformly distributed. In contrast, the CeO_x__B agglomerates (Figure 1d) were relatively large and locally distributed. The same pattern was also confirmed in the EDX mapping images of the Ce element (Figure 1b,c). Nevertheless, the average size of all the hybrid nanoparticles (Figure 1a,d) was almost similar at the level of 7–8 nm, irrespective of which dispersion was used. It was fairly different from the tendency in the average size of CeO_x__A and CeO_x__B nanoparticles dispersed in the aqueous N-propanol mixture. The statistical size analysis on the basis of the SEM images showed that the larger particle size in the dispersion was not due to aggregation of CeO_x__B nanoparticles but rather due to the swelling of 3M 725EW ionomer in contact with the nanoparticles.

Figure 2 exhibits the proton conductivity of the PFSA–PTFE reinforced membranes as a function of temperature. The proton conductivity in all the membranes increased at elevated temperatures because of the synergistic effect of the improved movement of 3M 725EW ionomer chains in the membrane matrices and the faster proton mobility. The proton conductivity values of PFM were slightly higher compared to those of the corresponding freestanding membrane (FSM). This was because the porous PTFE support film in PFM suppressed excessive membrane swelling, particularly in the planar direction, and increased the density of –SO_3_^−^ groups per unit volume. This difference in proton conductivity increased with increasing temperature. After CeO_x__A and CeO_x__B dispersions were added with the same cerium element content, the resulting CeO_x__A PFM and CeO_x__B PFM experienced serious losses in their proton conduction capabilities in comparison to the original PFM. This might be due to the presence of remaining Ce^3+^ ions in the reinforced membranes. Under an assumption that there was no additional CeO_x_ conversion, it was inferred that 65.6 and 30.9 mol % of the initially added Ce^3+^ ions would remain in CeO_x__A PFM and CeO_x__B PFM, respectively. The losses in their proton conduction increased proportionally with their remaining Ce^3+^ content. Thus, CeO_x__A PFM showed proton conduction lower than CeO_x__B PFM in the temperature range. In the meantime, the membrane acidification was effective enough to remove the remaining Ce^3+^ ions from CeO_x__A PFM and CeO_x__B PFM. The resulting H–CeO_x__A PFM and H–CeO_x__B PFM had proton conductivity values much higher than those of the corresponding reinforced membranes containing residual Ce^3+^ ions. In addition, their proton conductivity values were superior to those of PFM. The degree of improvement was more pronounced in H–CeO_x__B PFM with a relatively high CeO_x_ content. This indicated that the CeO_x_ hybrid nanoparticles containing –OH groups embedded in strongly acidic 3M ionomer matrices played a role in conducting protons.

Figure 3 displays the hydrogen permeability of PFSA–PTFE reinforced membranes in addition to FSM. The hydrogen permeation in all the membranes became fast as a function of temperature owing to the increase in hydrogen diffusivity. PFM had hydrogen permeability values lower than those of FSM in the temperature range. Considering that hydrogen permeability values of PTFE were inferior to those of 3M 725EW ionomers, this meant that severe defects or voids that might occur during reinforced membrane preparation were not formed. After the incorporation of CeO_x_ hybrid nanoparticles, the hydrogen permeability values of H–CeO_x__A PFM and H–CeO_x__B PFM slightly increased, regardless of their identities. It was inferred that the incomplete compatibility between CeO_x_ nanoparticles and 3M 725 ionomer matrices in the resulting reinforced membranes might have induced the formation of interfaces, which weakened the hydrogen barrier properties of the resulting reinforced membranes.

Figure 4 shows the concentration of fluoride ions that were products of the radical-induced chemical decomposition of the side chain and/or the main chain of 3M 725EW ionomer matrices in the reinforced membranes [48,49,50]. Here, the fluoride ions included F^−^ and CF_3_COO^−^ [51]. When Ce^3+^ ions or CeO_x_ hybrid nanoparticles were introduced as radical scavengers, the high fluoride ion concentration of PFM was significantly lowered. Their antioxidant effect, however, seemed to be quite different from each other, in spite of the use of the same cerium element content. The fluoride ion concentration obtained from Ce(NO_3_)_3_ PFM was 16–50% higher than those detected from the reinforced membranes containing CeO_x_ hybrid nanoparticles. This implies that CeO_x_ hybrid nanoparticles may be more effective than Ce^3+^ ions originated from Ce(NO_3_)_3_, in eliminating radicals. This trend may be associated with the radical scavenging reaction mechanism depending on the oxidation number of cerium ions. It is known that both Ce^3+^ and Ce^4+^ ions can participate in eliminating aggressive radicals (e.g., ·O_2_^−^) as radical scavenging catalysts following repeated chemical transformation from Ce^4+^ to Ce^3+^ and vice versa (see Equation (6) to (7) [52]). More specifically, when Ce^4+^ ions meet ·O_2_^−^ chosen as one of the aggressive radicals, Ce^4+^ ions can be easily converted to Ce^3+^ ions (Equation (6)). Meanwhile, Ce^3+^ ions can be altered to Ce^4+^ ions only with the help of acidic protons. Because Ce^3+^ ions in Ce(NO_3_)_3_ PFM are tightly ionically bound to –SO_3_^−^ groups of its acidic 3M 725EW ionomer matrices, the reduction reaction of the Ce^3+^ ions accompanied by the consumption of acidic protons would be certainly slow. It would make the chemical reaction of Equation (7) a rate-determining step in the radical removal mechanism. The slow reaction from Ce^3+^ to Ce^4+^ ions seems to be responsible for the weakened antioxidant effect in Ce(NO_3_)_3_ PFM.
**·**O_2_^−^ + Ce^4+^ → O_2_ + Ce^3+^(6)
**·**O_2_^−^ + Ce^3+^ + 2H^+^ → H_2_O_2_ + Ce^4+^(7)

This difference in the antioxidant effect between Ce^3+^ and CeO_x_ hybrid nanoparticles was also observed in CeO_x__A PFM and CeO_x__B PFM. Under the premise that highly dissociated Ce^3+^ contents measured via ICP analysis of CeO_x__B dispersion (Table 1) were lower than that in CeO_x__A dispersion and that the reduced content of the Ce^3+^ ions was used for the formation of additional CeO_x_ hybrid nanoparticles in CeO_x__B dispersion, CeO_x__B PFM had CeO_x_ hybrid nanoparticle content higher than CeO_x__A PFM. In addition to the influence of CeO_x_ content, it is worth paying attention to the identities of CeO_x_ hybrid nanoparticles. The CeO_x_ hybrid nanoparticles in CeO_x__A dispersion are expected to exist mainly in the form of Ce(OH)_3_ where the oxidation number of cerium ion is +3 (i.e., Ce^3+^), while the CeO_x_ nanoparticles in CeO_x__B dispersion are expected to have a relatively high content of Ce^4+^ ions, which are derived from Ce(OH)_4_ and/or CeO_2_. The high CeO_x_ content and relatively high portion of Ce^4+^ ions in the CeO_x_ nanoparticles contributed to the improved chemical resistance of CeO_x__B PFM compared to CeO_x__A PFM when exposed under radical decomposition condition.

Surprisingly, H–CeO_x__A PFM and H–CeO_x__B PFM obtained after the extraction of Ce^3+^ via membrane acidification had much improved resistance to the radical decomposition compared to their counterpart membranes CeO_x__A PFM and CeO_x__B PFM, respectively. This unexpected result might be due to the additional CeO_x_ conversion of the residual Ce^3+^ ions in CeO_x__A PFM and CeO_x__B PFM during the membrane acidification and the consequent increase in CeO_x_ content. It is estimated that the boiling H_2_SO_4_ treatment used for the membrane acidification was effective enough to transform the residual Ce^3+^ ions to additional CeO_x_ hybrid nanoparticles.

It is important to monitor changes in the electrochemical single cell performance of the reinforced membranes after incorporation of CeO_x_ hybrid nanoparticles. All the cell voltage values in Figure 5a decreased as a function of current density because of overpotentials occurring in activation polarization region (I), ohmic polarization region (II), and concentration polarization region (III), irrespective of which membrane was used. Here, the overpotential means the potential difference between theoretical (i.e., 1.23 V) and actual cell voltages. The cell voltage values in the activation region were affected by the hydrogen barrier properties of the membrane materials. Although there were small differences between the reinforced membranes, their OCV values in the activation polarization region were close to the theoretical cell voltage. This meant that there were no serious defects or voids in the reinforced membranes, which could lead to a rapid reduction in their cell performance during PEMFC operation. Unlike reinforced membranes employing ion-type radical scavengers (e.g., Ce(NO_3_)_3_ [53,54,55] and Ce(CH_3_CO_2_)_3_ [56]), critical potential drops in the ohmic polarization region were not observed in either H–CeO_x__A PFM or H–CeO_x__B PFM. Their electrochemical performance was rather enhanced when compared to radical scavenger-free PFM. The potential drop in the region would depend on proton conduction across the PFSA ionomer located in the whole MEA from anode to cathode through the PEM as well as the contact resistance between the PEM and electrodes. Because all the MEA components were identical and the same MEA fabrication method was applied, this improvement in their electrochemical performance was assumed to be due to proton conductivity being enhanced in H–CeO_x__A PFM and H–CeO_x__B PFM, with CeO_x_ hybrid nanoparticles acting as additional proton conductors (see Figure 2). H–CeO_x__B PFM with a relatively high CeO_x_ content showed current–voltage polarization curve little bit higher than that of H–CeO_x__A PFM in the region. The single cell performance was improved further as applied humidity (see Figure 5b) and pressure (Figure 5c) increased. The maximum cell performance was obtained when a pressure of 2 bar_g_ and a humidity of 100% RH were applied (see Figure 5d). The electrochemical performance trends between the reinforced membranes were, however, not changed.

Figure 6 exhibits the electrochemical effectiveness of CeO_x_ hybrid nanoparticles as powerful radical scavengers to extend the service life of the resulting PFSA–PTFE reinforced membranes. As the accelerated test to evaluate chemical stability progressed (Figure 6a), the OCV value of PFM without the CeO_x_ hybrid nanoparticles decreased rapidly up to 20% or more of its initial OCV value within 100 h, showing a very fast irreversible loss rate of 50.2 mV/day. This situation was more severe when its electrochemical hydrogen crossover was observed as a function of time (Figure 6b). The hydrogen crossover in the initial test was stable enough to maintain 1.91 mA/cm^2^ in the measurement voltage range, but the level after 100 h increased up to 75 to 175 mA/cm^2^, which far exceeded the hydrogen crossover cutoff of 15 mA/cm^2^. On the other hand, the service life could be extended by incorporating CeO_x_ hybrid nanoparticles into the resulting reinforced membranes. The content and/or identities of the CeO_x_ hybrid nanoparticles still seemed to play a key role in determining the service life of PFSA–PTFE reinforced membranes. Until 160 h, H–CeO_x__A PFM maintained its OCV value within 20% of its initial value, but its hydrogen crossover was beyond the cutoff. In the case of H–CeOx_B PFM, expected to contain a relatively high amount of CeO_x_ nanoparticles based on Ce^4+^ ions, the improvement in chemical durability was very noticeable. Even after 200 h or more, its irreversible loss did not exceed 9% of the initial value, and the hydrogen crossover was still below the cutoff. The interfacial issue observed in the hydrogen permeability test (Figure 3) did not affect the electrochemical hydrogen crossover. This can be explained as originating from the difference in their measurement environments; the pressure difference on both sides of the sample membrane was 1 and 0 bar_g_ in the hydrogen permeability test and LSV test, respectively. Considering that the theoretically calculated cerium element content in H–CeO_x__B PFM was at a relatively low level (i.e., 1.7 mg/cm^3^ in a PFSA matrix or 2.5 µg/cm^2^ on the basis of a reinforced membrane with a thickness of 15 µm), its electrochemical effect on the chemical durability is meaningful.

## 4. Conclusions

CeO_x_ hybrid nanoparticles with an average size of 7–8 nm in solid state were successfully synthesized from Ce(NO_3_)_3_ through in situ stepwise conversion reactions composed of HCl-catalyzed hydrolysis reaction and NH_3_OH-catalyzed condensation reaction in an aqueous N-propanol mixture where 3M 725EW ionomer particles were homogeneously distributed. Here, the 3M 725EW ionomer was used as a polymeric dispersant to prevent spontaneous aggregation and subsequent precipitation of CeO_x_ nanoparticles chemically converted in the aqueous medium by making multivalent Ce^3+^ ions maintain ionic bonding with the –SO_3_^−^ groups. As the stepwise reactions from acid-catalyzed to base-catalyzed reaction proceeded, the initial Ce^3+^ ion content decreased and the yield of CeO_x_ conversion increased from 34.4 to 69.1 mol %. The resulting CeO_x__A and CeO_x__B dispersion maintained stable dispersion phases without any aggregation and precipitation, even in long-term storage for months.

When incorporated as radical scavengers during reinforced membrane formation, CeO_x_ hybrid nanoparticles contributed to improved proton conductivity of H–CeO_x__A PFM and H–CeO_x__B PFM in comparison to that of CeO_x_-free reinforced membrane (i.e., PFM). This is attributable to the presence of –OH groups on the CeO_x_ hybrid nanoparticles acting as proton conductors within the acidic 3M 725EW ionomer matrices. The enhancement in their proton conduction capability compensated their potential drops in ohmic polarization region of the current–voltage polarization curves and instead led to improved single cell performance. These aspects were fairly different from those frequently observed in reinforced membranes containing Ce(NO_3_)_3_ as a radical scavenger. Moreover, there were no big changes in their hydrogen permeability and initial electrochemical hydrogen crossover levels following the addition of CeO_x_ hybrid nanoparticles. The weak resistance of PFM to free radical attacks was able to be improved to high levels by introducing CeO_x_ hybrid nanoparticles. The increment in radical resistance seemed to depend on both the content of CeO_x_ hybrid nanoparticles and the relative portion of Ce^4+^ ions in the nanoparticles. H–CeO_x__B PFM, with the highest content of CeO_x_ hybrid nanoparticles where the oxidation number of cerium is mainly +4, had the lowest concentration of fluoride ions detected in a Fenton solution, which has been conventionally used to evaluate the radical-induced chemical decomposition of PFSA ionomers. Surprisingly, the fluoride ion concentration was less than half the level detected in Ce(NO_3_)_3_ PFM for comparison. The electrochemical effectiveness of CeO_x_ hybrid nanoparticles as radical scavengers was validated using the OCV holding test. The OCV values of H–CeO_x__B PFM was continuously kept, showing the lowest irreversible loss and electrochemical hydrogen crossover levels over time.

From the perspective of both electrochemical performance and chemical durability, the strategy of applying CeO_x_ hybrid nanoparticles as radical scavengers, instead of Ce(NO_3_)_3_ or conventional CeO_2_ nanoparticles, appears to be very useful. The CeO_x_ hybrid nanoparticles developed in this study are expected to provide an effective avenue in designing ultimate radical scavengers and applying them. Finally, when highly purified CeO_x_ nanoparticles are applied for PEMs and electrodes, their impact on both electrochemical performance and durability will be addressed in separate publications in membrane- and electrochemistry-oriented journals.

## Figures and Tables

**Figure 1 membranes-11-00143-f001:**
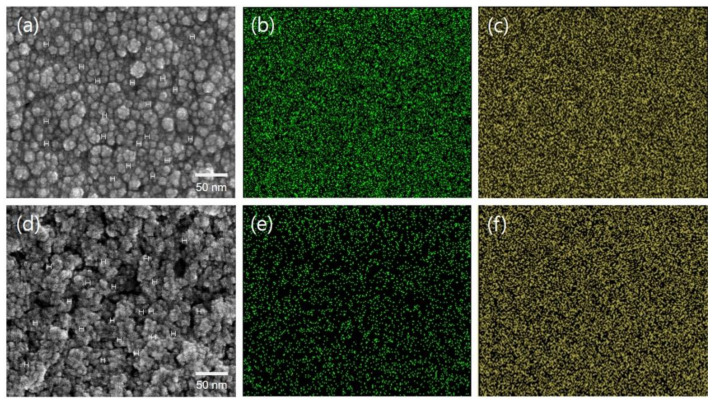
(**a**,**d**) FE-SEM and (**b**,**c**,**e**,**f**) EDX mapping images ((**b**,**e**) cerium and (**c**,**f**) oxygen element) of (**a**–**c**) CeO_x__A–3M ionomer hybrid nanoparticles and (**d**–**f**) CeO_x__B–3M ionomer hybrid nanoparticles in the solid state obtained after drying each dispersion.

**Figure 2 membranes-11-00143-f002:**
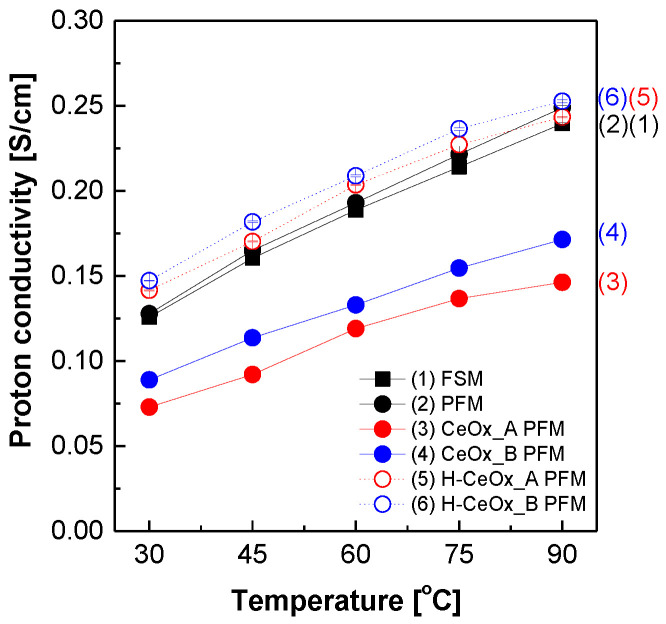
Proton conductivity of perfluorinated sulfonic acid–poly(tetrafluoroethylene) (PFSA–PTFE) reinforced membranes in deionized water. The freestanding membrane (FSM) was made up of 3M 725EW ionomer.

**Figure 3 membranes-11-00143-f003:**
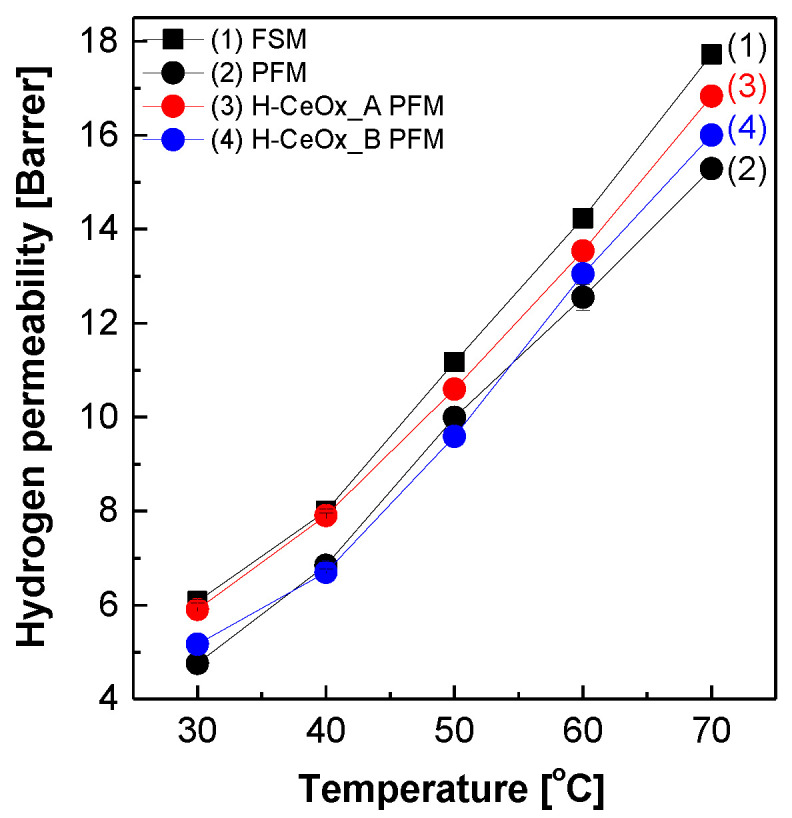
The hydrogen permeability of PFSA–PTFE reinforced membranes measured via time-lag method.

**Figure 4 membranes-11-00143-f004:**
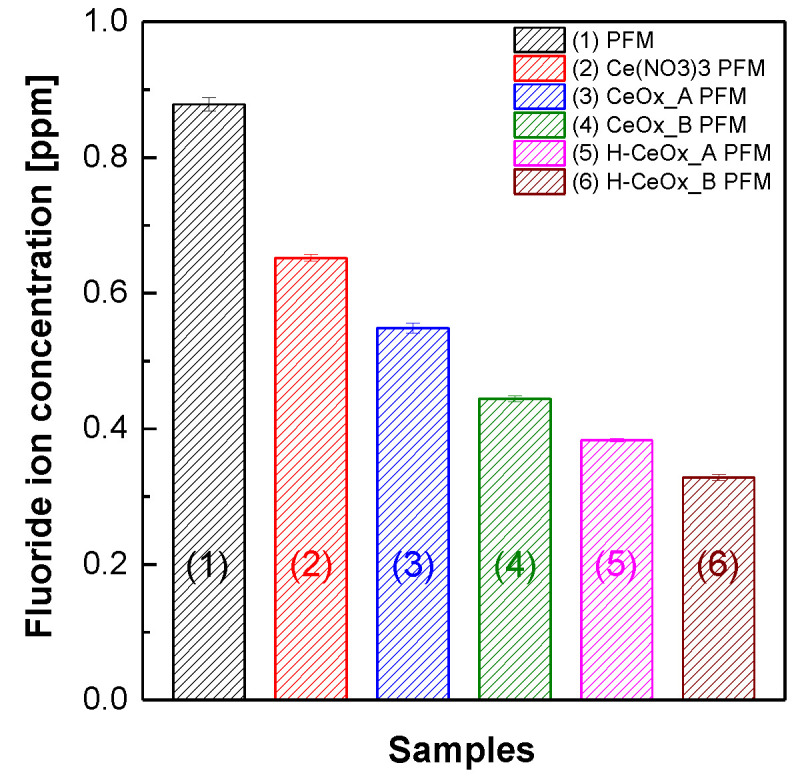
The concentration of fluoride ions detected after the chemical degradation of PFSA–PTFE reinforced membranes in a Fenton solution at 80 °C for 2 h.

**Figure 5 membranes-11-00143-f005:**
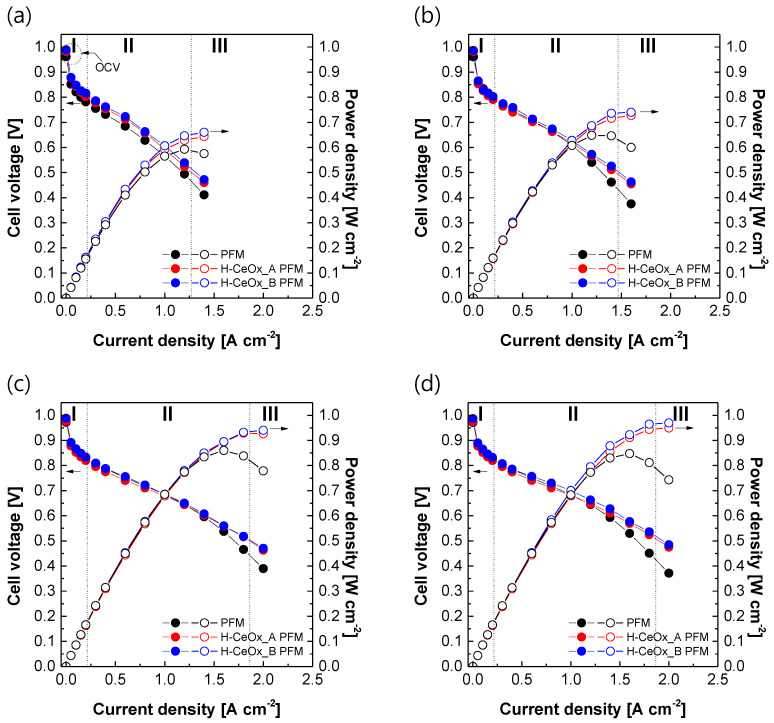
The current–voltage polarization curves of PFSA–PTFE reinforced membranes obtained at 65 °C according to changes in applied pressure ((**a**,**b**) 1.0 bar_g_ and (**c**,**d**) 2.0 bar_g_) and humidity ((**a**,**c**) 50% relative humidity (RH) and (**b**,**d**) 100% RH) values.

**Figure 6 membranes-11-00143-f006:**
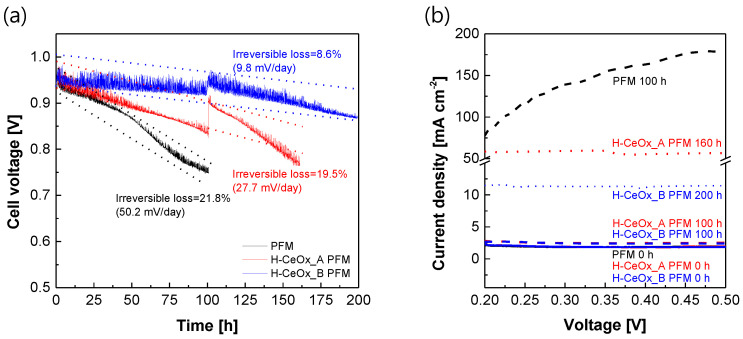
(**a**) Open-circuit voltage (OCV) values of PFSA–PTFE reinforced membranes as a function of operation time at 90 °C and 30% RH under the pressure of 1.5 bar_g_. (**b**) Hydrogen crossover obtained with the interval of 100 h via linear sweep voltammetry (LSV) analysis, which was conducted at 65 °C and 50% RH under the pressure of 1.0 bar_g_.

**Table 1 membranes-11-00143-t001:** Dispersion characteristics depending on CeO_x_ conversion reaction conditions.

Sample	Average Particle Size [nm]	Ce^3+^ Concentration [ppm] *^d^*			CeO_x_ Conversion *^f^* [mol%]	
		t *^e^* = 0	t *^e^* = 12	t *^e^* = 24	t *^e^* = 12	t *^e^* = 24
Pristine dispersion *^a^*	2.2 ± 0.2	0	0	0	0	0
Ce(NO_3_)_3_ dispersion	2.8 ± 0.3 *^b^*	15,200 *^b^*	–	–	–	–
CeO_x__A dispersion	61 ± 2 *^c^*	15,200 *^b^*	13,700	10,000	9.87	34.2
CeO_x__B dispersion	152 ± 5 *^c^*	15,200 *^b^*	9700	4700	36.2	69.1

*^a^* 5 wt % aqueous dispersion of 3M 725EW ionomer without any CeO_x_ nanoparticles; *^b^* measured immediately after the Ce(NO_3_)_3_ dissolution; *^c^* measured using each dispersion obtained after the chemical reaction for 24 h; *^d^* measured via inductively coupled plasma optical emission spectroscopy; *^e^* reaction time in hours; *^f^* CeO_x_ conversion = [(Ce^3+^ concentration at t = 0 – Ce^3+^ concentration at t = 12 or 24)/(Ce^3+^ concentration at t = 0)] × 100.

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
