# Peer review of "Highly Dispersed CeOx Hybrid Nanoparticles for Perfluorinated Sulfonic Acid Ionomer–Poly(tetrafluoethylene) Reinforced Membranes with Improved Service Life"

_membranes, 2021, doi:10.3390/membranes11020143_

Round 1
Reviewer 1 Report
Please see attached.

Reviewer 2 Report
- The introduction is slightly off topic. Why do the authors focus on the hydrogen electric vehicles without proposing in their conclusions their research contribution towards that? The introduction part needs to be rewritten or the authors need to include a section in the discussion part on how their research contributes to the topic that mentioned in the introduction.
- Page 2, line 50: “includes” needs to be corrected to “include”.
- Figure 1. What is the percentage distribution of the elements analyzed and how was the dispersion affected it during different nanoparticles?
- Figure 2. Do the authors expect the proton conductivity to reach a plateau after a certain temperature in their novel membranes?
- Figure 3. In general, when membranes have not perfect selectivity, high permeance contributes to higher losses hence it might not be the goal. How do the authors explain a change in the H2 permeability behavior of the H-CeOx_A PFM and H-CeOx_B PFM? How does temperature affect the trend adversely? Also, this graph is too busy to provide for a clear assessment. The authors might reconsider to alter it (either split it into two or make it more visible).
